# Predicting Individual Tree Diameter of Larch (*Larix olgensis*) from UAV-LiDAR Data Using Six Different Algorithms

**Yusen Sun [1], Xingji Jin [1,*,†], Timo Pukkala [1,2,†] and Fengri Li [1]**

[1] Key Laboratory of Sustainable Forest Ecosystem Management—Ministry of Education, School of Forestry, Northeast Forestry University, Harbin 150040, China; sunyusen@nefu.edu.cn (Y.S.); timo.pukkala@uef.fi (T.P.); fengrili@nefu.edu.cn (F.L.)

[2] School of Forest Sciences, University of Eastern Finland, P.O. Box 111, 80101 Joensuu, Finland

* Correspondence: xingjijin@nefu.edu.cn; Tel.: +86-451-8219-0878

† These authors contributed equally to this work.

**Abstract:** Individual tree detection is an increasing trend in LiDAR-based forest inventories. The locations, heights, and crown areas of the detected trees can be estimated rather directly from the LiDAR data by using the LiDAR-based canopy height model and segmentation methods to delineate the tree crowns. However, the most important tree variable is the diameter of the tree stem at the breast height (DBH) which can seldom be interpreted directly from the LiDAR data. Therefore, the use of individually detected trees in forest planning calculations requires predictions for the DBH. This study tested six methods for predicting the DBH from laser scanning data collected by an unmanned aerial vehicle from *Larix olgensis* plantations located in northeast China. The tested methods were the linear regression model (LM), a linear model with ridge regularization (LMR), support vector regression (SVR), random forest (RF), artificial neural network (ANN), and the k-nearest neighbors (KNN) method. Both tree-level and stand-level metrics derived from the LiDAR point cloud data (for instance percentiles of the height distribution of the echoes) were used as potential predictors of DBH. Compared to the LM, all other methods improved the accuracy of the predictions. On the other hand, all methods tended to underestimate the DBH of the largest trees, which could be due to the inability of the methods to sufficiently describe nonlinear relationships unless different transformations of the LiDAR metrics are used as predictors. The support vector regression was evaluated to be the best method for predicting individual tree diameters from LiDAR data. The benefits of the methods tested in this study can be expected to be the highest in the case of little prior knowledge on the relationships between the predicted variable and predictors, a high number of potential predictors, and strong mutual correlations among the potential predictors.

**Keywords:** random forest; artificial neural network; k-nearest neighbors; support vector regression; ridge regulation; machine learning

## 1. Introduction

Forest ecosystems play an important role in maintaining ecological balance and carbon cycle, regulating local and regional climate, and preserving biosphere stability [1]. Information on tree cover is required for the management of forest ecosystems and to support policies on ecological restoration and climate change mitigation [2]. Traditional forest inventories often employ intensive field samplings with accurate measurements in sample plots [3]. However, field-based forest inventories are labor-intensive and time-consuming, and therefore expensive for collecting data from large areas.

Remote sensing is an effective tool for monitoring wide forest areas [4]. As an active remote sensing technology, airborne light detection and ranging (LiDAR) can directly capture detailed information on forest canopies in three dimensions from large areas [2,5]. LiDAR has proven to have a high potential in predicting forest attributes and acquiring auxiliary information for sampling inventories [6].

However, the flying cost per hectare of airborne LiDAR inventory may be high if the inventory area is not large. Airborne LiDAR inventories may also be limited by adverse flight conditions [7]. Compared with airborne LiDAR, laser scanning from unmanned aerial vehicles (UAVLS) has the advantage of low material and operational costs and high flexibility [8]. UAVLS has been used successfully in several recent studies to predict the diameter distribution of trees [9], mean tree height [8], and aboveground carbon stock [10]. Furthermore, due to the high point density of UAVLS data, the crowns of individual trees can be delineated, improving the accuracy of individual tree detection (ITD) [11].

The diameter at breast height (DBH) correlates strongly with other tree attributes and is easy to measure accurately in the field [12]. It is the most common predictor variable in stem volume equations, tree growth models, and biomass equations. Several studies have been recently carried out where LiDAR data were applied to estimate the DBH of individual trees [2,8,12].

UAVLS data may not be sufficient for the direct detection of the stems of individual trees, due for instance to canopy obstruction and non-optimal scanning angle [13]. However, tree diameter can be predicted through case-specific models that account for the strong relationship between DBH and LiDAR-derived canopy height [12]. Estimates of tree locations and diameters over large continuous forest areas would make it possible to optimize forest management at the level of individual trees [14–16].

The existing studies on the estimation of DBH from ALS data mostly employ tree-level LiDAR-derived metrics [17]. However, the relationship between LiDAR metrics and DBH may vary between sites and stands [18]. For example, the diameter-height curve of an even-aged stand of shade-intolerant species is often higher in older stands, i.e., a tree of a given DBH is taller the older the stand is [19]. Therefore, it is logical to test stand-level LiDAR-derived metrics in the prediction of DBH, although it would greatly increase the number of potential predictors. Multicollinearity between different LiDAR-derived metrics is another potential problem when both tree- and stand-level LiDAR metrics are used in model development.

The traditional statistical model-fitting methods, such as the ordinary least squares (OLS) regression, require large sample sizes in the presence of multicollinearity. In the case of multicollinearity, the variances of regression coefficients are large, which makes it difficult to test hypotheses concerning the effects of predictors [20]. The assumptions of OLS regression include linearity, independence of predictors variables, normality of the distribution of residuals, and homogeneity of variance [20].

Nonparametric machine learning techniques, such as random forest (RF), support vector regression (SVR), k-nearest neighbors imputation (KNN), and artificial neural networks (ANN) are alternatives to the traditional regression analysis, especially in the case of a high number of potential predictors, high degree of multicollinearity, and little prior knowledge on the relationships between variables [21]. The use of ridge or lasso regularization in regression analysis also mitigates some of the problems of OLS since it increases the robustness of the model [22].

Nonparametric approaches have already been used in the prediction of forest attributes from LiDAR metrics. For example, Corte et al. (2020) used the support vector regression approach, which showed similar predictive performance as OLS for modeling diameter, height, and volume [17]. Hao et al. (2021) used the random forest approach to estimate the individual tree diameter [11]. Pascual et al. (2019) used the random forest approach to estimate forest attributes [23]. However, more research is needed on different procedures for modeling DBH and other tree attributes with LiDAR metrics.

In this study, we compared six alternative methods for predicting individual tree diameter from UAVLS data in northeast China. The tested methods were linear regression model (LM), linear model with ridge regularization (LMR), support vector regression (SVR), random forest (RF), artificial neural network (ANN), and the k-nearest neighbors method (KNN). For this purpose, we (1) calculated a large number of tree-level and stand-level LiDAR metrics for trees detected from UAVLS data, (2) used the recursive feature

elimination (RFE) method to deal with the high data dimensionality, (3) determined the optimal hyper-parameter values for each method, and (4) compared the performance of the algorithms in an independent validation dataset. Our comprehensive analyses reveal the benefits and potential pitfalls of alternative prediction methods and help to understand the mechanisms of predicting DBH from LiDAR data.

## 2. Materials and Methods

### 2.1. Study Sites and Field Data

The study area is located at the Mengjiagang forest farm (45°30′16″–46°20′20″N, 130°32′0″–130°52′6″E) of Huanan country in Heilongjiang province of northeast China (Figure 1). The total forest area of the farm is 15,503 ha, of which 4438 ha are natural forests and the rest are plantations. The average elevation of the case study forest is 250 m above sea level [16]. The plantation forests are dominated by coniferous tree species, predominantly *Pinus koraiensis*, *Pinus sylvestris*, *Larix olgensis*, and *Picea asperata*.

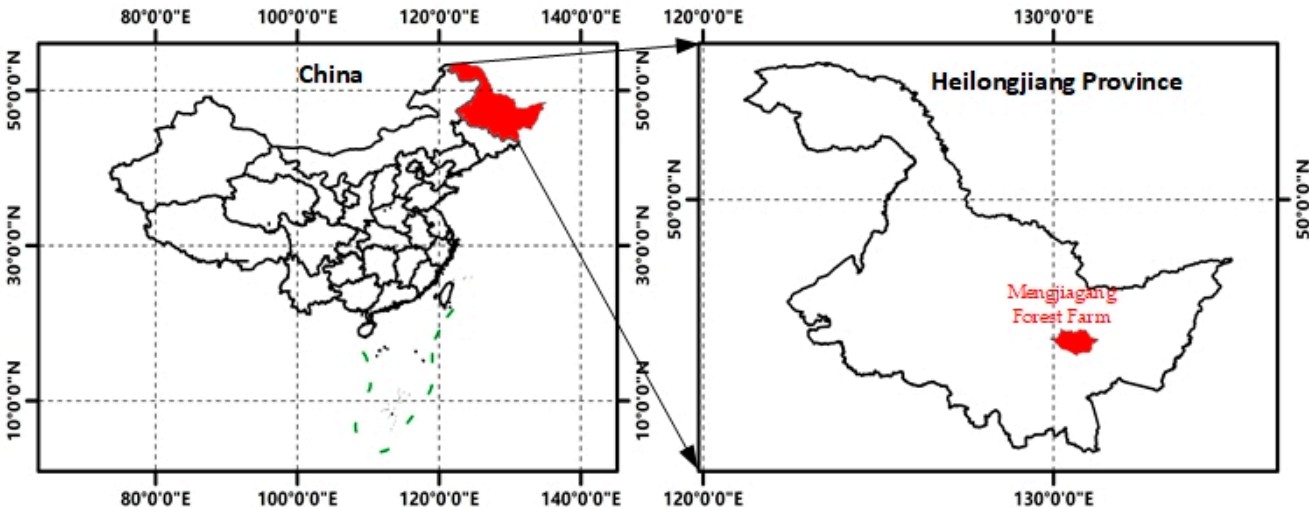

**Figure 1.** Map of the Mengjiagang forest farm in Heilongjiang province, northeast China, showing the location of the study area.

Larch (*Larix olgensis* A. Henry) is one of the main economic tree species in Northeast China. It accounts for 31% (4800 ha) of the tree plantations of the Mengjiagang forest farm. The breast height diameter of the trees of the current plantations ranges from 5 to 40 cm and the age of the planted stands ranges from 12 to 52 years. Most of the larch plantations have been planted using a density of 3300 stems/ha. The stands are thinned regularly so that a typical density of a near-mature stand is 500 stems/ha [16].

This study developed models for the diameter at breast height (DBH) of larch. A total of 53 square-shaped sample plots (30 m × 30 m) were established shortly after the LiDAR flights were conducted in July 2019. The field-measured inventory data included DBH, total tree height, crown width, and tree position. A total of 4109 trees at least 5 cm in DBH were measured for the DBH (at the height of 1.3 m from the ground) using diameter tape. The total height of the tree was measured using a Vertex IV instrument (Haglöfs, Sweden). Crown width was measured in four directions by using a tape measure. The positions of the trees were measured using a real-time kinetic (RTK) global navigation satellite system (GNSS) (UniStrong G10A, Beijing, China) for which the positioning errors were less than 1 m. Summary statistics of the tree variables are presented in Table 1.

**Table 1.** Descriptive statistics of forest measurements.

| Variable | Mean | Std Dev | Minimum | Maximum |
|---|---|---|---|---|
| DBH (cm) | 17.39 | 7.80 | 5.00 | 38.80 |
| Total height (m) | 15.61 | 6.69 | 5.20 | 32.70 |

Note: Std Dev = Standard deviation.

### 2.2. UAVLS Data

UAVLS data were collected with a RIEGL mini VUX-1UAV LiDAR scanner (RIEGL Laser Measurement Systems GmbH, Horn, Austria) on 12 August 2019. The scanner was carried by a DJI M600 Pro unmanned aerial vehicle. The collected raw data were processed with the software package RiPROCESS (http://www.riegl.com/products/software-packages/riprocess/, accessed on 8 October 2019) to generate UAVLS point cloud data [24]. The total scanned area was 669 ha, consisting of three sub-areas, each of which contained a rectangular survey area of 1 km × 1.5 km. The total scanning time was about one hour. The average point density was 136 pulses/m$^2$. The flight speed was 10 m/s and the flight altitude was 180 m above ground. The field of view (FOV) of the laser scanner was 330°. The operational scanning specifications are summarized in Table 2.

**Table 2.** Details of the operational parameters for the UAVLS data collection.

| Variable | Value |
|---|---|
| Laser pulse repetition rate | 380 kHz |
| Accuracy | 5 mm |
| Maximum echo number | 5 |
| Maximum scan speed | 200 scans/second |
| Echo signal intensity | 16 bit |
| Laser wavelength | 1550 nm |
| Beam divergence | 0.5 mrad |

### 2.3. UAVLS Data Pre-Processing

The whole process of developing the prediction models for DBH is shown in Figure 2. After pre-processing, both stands and the crowns of individual trees were delineated using the UAVLS data, after which a high number of metrics were calculated for the stands and trees.

As the first step of data pre-processing, the Gaussian-smoothing filter was employed to remove the noise data points by removing small variations from the canopy surface. The degree of smoothness was determined by the standard deviation [25].

Then, cloth simulation filtering was used to separate non-ground and ground point clouds using 0.5 as the value of the grid resolution parameter, 0.6 as the time step, and 3 as the rigidness parameter [26]. Kriging interpolation was used to generate a digital elevation model (DEM) from the ground points in 1-m spatial resolution using the ArcGIS 10.6 software [27].

The point cloud data were height-normalized by using the DEM. A canopy height model (CHM) was created from the normalized point cloud using the GridSurfaceCreate function in the LiDAR360 software (www.lidar360.com, accessed on 8 October 2019) [2]. The CHM was subsequently used for individual tree delineation. Based on Hao et al. (2021), the resolution of the CHM was set to 0.1 m, which made it possible to detect tree crowns wider than 0.1 m [11].

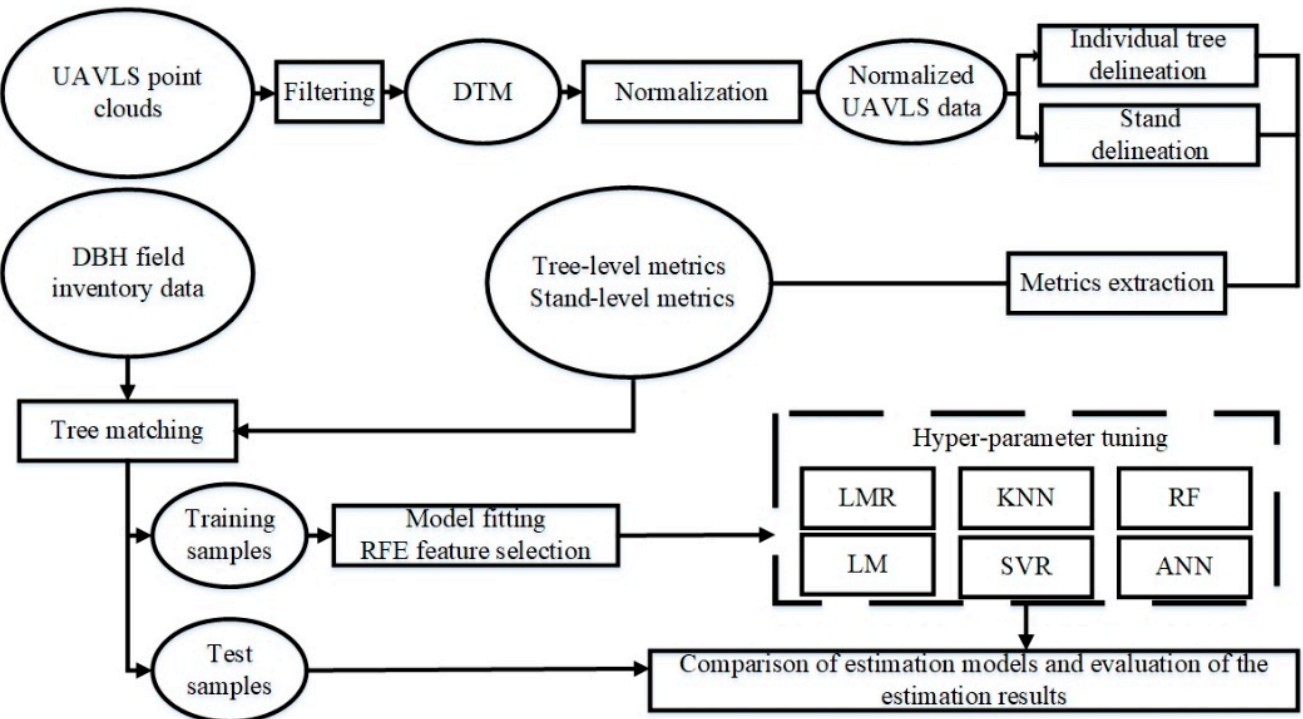

**Figure 2.** An overview of the workflow for DBH estimation by different methods using UAVLS data.

### 2.4. Individual Tree Delineation

The first step in the tree-level LiDAR metrics extraction was the delineation of individual trees from the CHM. It was done with the watershed segmentation algorithm of the LiDAR360 software, using 90 as the buffer size and 0.8 as the crown base height threshold. The algorithm was used as explained in Chen et al. (2006) [28].

The algorithm segmented the CHM into different polygons, polygon boundaries representing the perimeters of individual tree crowns (Figure 3). The maximum elevation of each polygon indicated the treetop. Those LiDAR-detected trees that could be matched with field-measured trees were used to develop models for DBH. Those segmented trees that could not be matched with field-observed trees were either trees that failed to produce local maxima in the CHM, or they were local maxima that did not represent treetops (Figure 3). In total, 79% of the trees were detected with commission errors in 22% of the trees. Out of 4109 field-measured trees, 3815 trees were correctly matched and used in modeling.

### 2.5. Stand Delineation

The stands were delineated using the method of Sun et al. (2021) [16]. The method employs simulated annealing (SA) to maximize an objective function where the criteria are small within-stand variation in the UAVLS attributes (weight 0.7), sufficient stand area (weight 0.15), and roundish shape of the stands (weight 0.15). The result depends on the weights of the criteria, but the usual aim is to avoid delineations with a high number of very small stands or irregular stand shapes.

In this study, the 95% percentiles of the height distribution of the echoes with 1-m$^2$ raster cells were used as input data in the stand delineation (Figure 4). A detailed description of the algorithm and its parameters is available in Sun et al. (2021) [16]. The stand delineation obtained from the SA method explained 80% of the variation in canopy height. The average stand area was 1.2 ha. All trees that were within the same stand received the same values of stand level metrics.

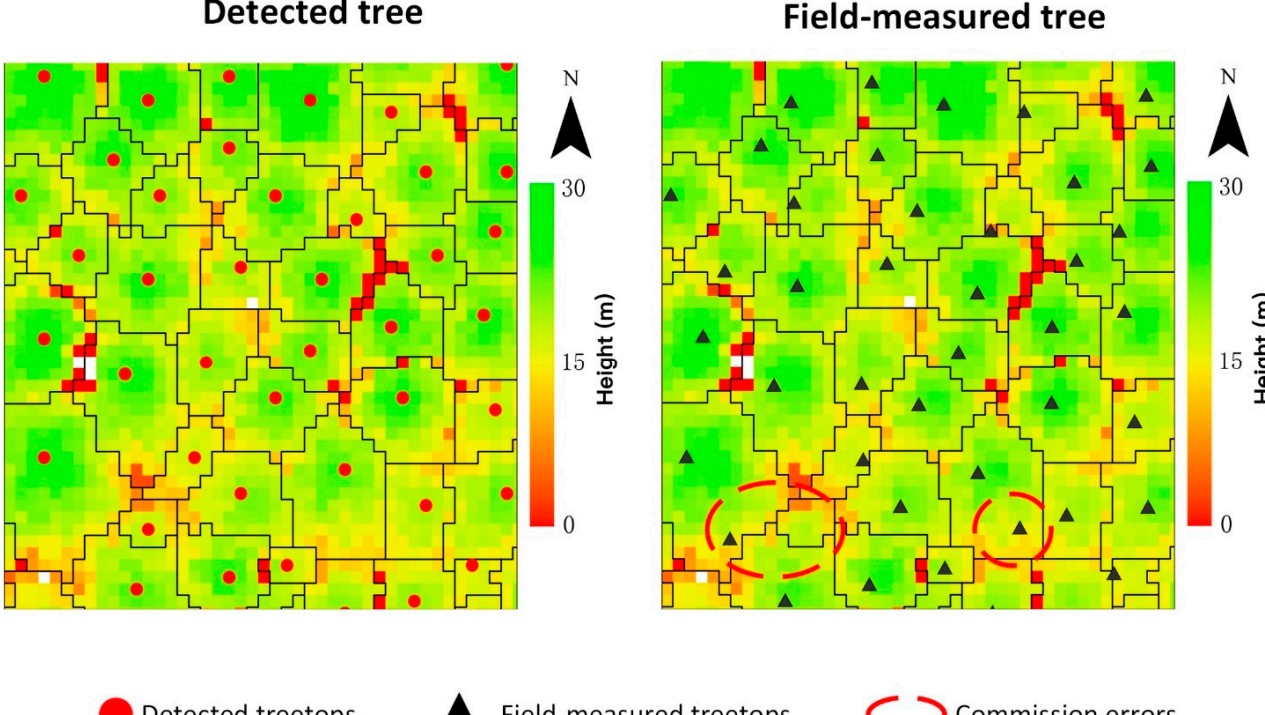

**Figure 3.** Examples of individual tree delineation. Polygon boundaries represent individual tree crowns. White cells represent no data cells.

### 2.6. Calculation of the LiDAR Metrics

The normalized point clouds and the delineations of trees and stands were used to calculate 82 tree- and stand-level LiDAR metrics that were used as potential predictors of DBH (Table 3).

The maximum echo height within the perimeters of an individual tree may be used to predict tree height. The coefficient of determination ($R^2$) for the linear regression between the LiDAR-derived and field-measured height was 0.8834. However, predicted tree heights were not used to predict DBH. The following categories of metrics were computed: density-related metrics ($D_{10}$, $D_{20}$, ..., $D_{90}$, and *SD* in Table 3); canopy volume metrics (*CPA*); topography-related metrics (*Slope* and *Aspect*); and height-related metrics (all other variables listed in Table 3).

**Table 3.** Summary of the tree- and stand-level metrics derived from UAVLS data that were used for DBH estimation.

| Abbreviation | Description |
| --- | --- |
| Tree level metrics | |
| $H_5{}^T$, $H_{10}{}^T$, $H_{20}{}^T$, $H_{25}{}^T$, $H_{30}{}^T$, $H_{40}{}^T$, $H_{50}{}^T$, $H_{60}{}^T$, $H_{75}{}^T$, $H_{80}{}^T$, $H_{90}{}^T$, $H_{95}{}^T$, $H_{99}{}^T$ | Percentiles of the height distributions of the echoes within the perimeters of individual trees (5th, 10th, ..., 95th, and 99th). |
| $H_{max}{}^T$, $H_{min}{}^T$, $H_{mean}{}^T$, $H_{med}{}^T$ | Maximum, minimum, mean, and medium of the echo heights within the perimeters of individual trees. |
| $H_{var}{}^T$, $H_{std}{}^T$, $H_{cv}{}^T$, $H_{ske}{}^T$, $H_{kur}{}^T$ | Variance, standard deviation, coefficient of variation, skewness, and kurtosis of the echo heights within the perimeters of individual trees. |

**Table 3.** *Cont.*

| Abbreviation | Description |
|---|---|
| $H_{IQ}{}^{T}$ | $H_{75}{}^{T}-H_{25}{}^{T}$ |
| $AIH_5{}^{T}, AIH_{10}{}^{T}, AIH_{20}{}^{T}, AIH_{25}{}^{T}, AIH_{30}{}^{T},$ $AIH_{40}{}^{T}, AIH_{50}{}^{T}, AIH_{60}{}^{T}, AIH_{75}{}^{T}, AIH_{80}{}^{T},$ $AIH_{90}{}^{T}, AIH_{95}{}^{T}, AIH_{99}{}^{T}$ | Percentiles of accumulated echo heights within individual tree perimeters (5th, 10th, . . . , 95th, and 99th). |
| $RCH_{max}, RCH_{mean}$ | The ratios of a target tree's height to the maximum and mean tree height of the stand. |
| $D_{10}{}^{T}, D_{20}{}^{T}, D_{30}{}^{T}, D_{40}{}^{T}, D_{50}{}^{T}, D_{60}{}^{T}, D_{70}{}^{T},$ $D_{80}{}^{T}, D_{90}{}^{T}$ | The proportion of points above the quantiles (10th, 20th, . . . , 80th, and 90th) of the total number of points within the perimeters of a tree. |
| $CPA$ | The projected area of the tree crown. |
| Stand level metrics | |
| $H_5{}^{S}, H_{10}{}^{S}, H_{20}{}^{S}, H_{25}{}^{S}, H_{30}{}^{S}, H_{40}{}^{S}, H_{50}{}^{S},$ $H_{60}{}^{S}, H_{75}{}^{S}, H_{80}{}^{S}, H_{90}{}^{S}, H_{95}{}^{S}, H_{99}{}^{S}$ | Percentiles of the echo height distributions (5th, 10th, . . . , 95th, and 99th) within a stand. |
| $H_{max}{}^{S}, H_{min}{}^{S}, H_{mean}{}^{S}, H_{med}{}^{S}$ | Maximum, minimum, mean and medium values of the echo heights within a stand. |
| $H_{var}{}^{S}, H_{std}{}^{S}, H_{cv}{}^{S}, H_{ske}{}^{S}, H_{kur}{}^{S}$ | Variance, standard deviation, coefficient of variation, skewness, and kurtosis of the echo heights within a stand. |
| $H_{IQ}{}^{S}$ | $H_{75}{}^{S}-H_{25}{}^{S}$ |
| $AIH_5{}^{S}, AIH_{10}{}^{S}, AIH_{20}{}^{S}, AIH_{25}{}^{S}, AIH_{30}{}^{S},$ $AIH_{40}{}^{S}, AIH_{50}{}^{S}, AIH_{60}{}^{S}, AIH_{75}{}^{S}, AIH_{80}{}^{S},$ $AIH_{90}{}^{S}, AIH_{95}{}^{S}, AIH_{99}{}^{S}$ | Percentiles of accumulated echo heights (5th, 10th, . . . , 95th, and 99th) within a stand. |
| $D_{10}{}^{S}, D_{20}{}^{S}, D_{30}{}^{S}, D_{40}{}^{S}, D_{50}{}^{S}, D_{60}{}^{S}, D_{70}{}^{S},$ $D_{80}{}^{S}, D_{90}{}^{S}$ | The proportion of points above the quantiles (10th, 20th, . . . , 80th, and 90th) of the total number of points within a stand. |
| $SD$ | Stem density measured from individual tree segmentation results within a stand. |
| *Slope* | The average slope of the stand (degrees). |
| *Aspect* | The aspect of the stand, calculated as the azimuth of an inclined plane (degrees). |

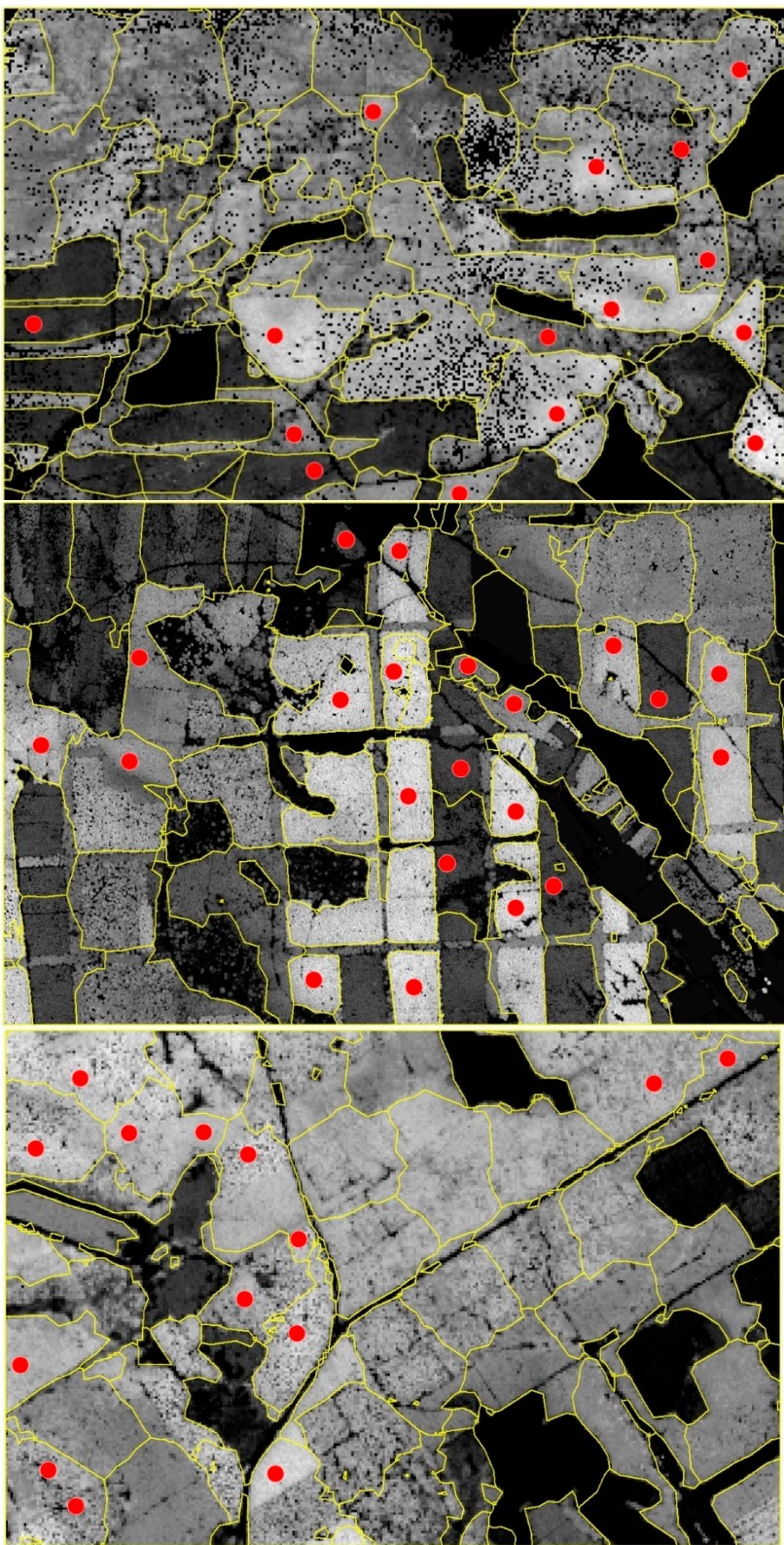

**Figure 4.** Stand delineation overlaid with the 95% percentile of the height distribution of the echoes (1 km × 1.5 km). A lighter tone indicates a higher value of the attribute. Red points represent sample plots and yellow lines are stand boundaries.

### 2.7. Modeling Methods

Six methods, namely linear model (LM), linear model with ridge regularization (LMR), support vector regression (SVR), k-nearest neighbors imputation (KNN), random forest (RF), and artificial neural networks (ANN), were used for DBH modeling.

Preliminary model fittings showed that $H_{max}{}^T$ and CPA (see Table 3 for the explanation of these abbreviations) were among the most important predictors of DBH. $H_{max}{}^T$ represents LiDAR-estimated tree height and CPA represents LiDAR-estimated projection area of the tree crown. Since the shape of the relationship between tree DBH and tree height is known to be nonlinear [19], $H_{max}{}^T$ was transformed into $(H_{max}{}^T)^2$ to linearize the relationship. In the same way, since DBH is a one-dimensional tree attribute and CPA is two-dimensional, the square root of CPA was used to linearize the relationship between CPA and DBH. These transformations, which were used as additional predictors compared to Table 3, allowed fair comparisons between the modeling methods, of which some have been developed only for linear relationships (especially LM and LMR).

All algorithms were implemented in the Python 'sklearn' (scikt-learn) package [29]. The LM is a parametric method that can be expressed as

$$y = a_0 + a_1 x_1 + a_2 x_2 + \ldots a_n x_n + \varepsilon \tag{1}$$

where $a_0$ is the intercept, $a_1, a_2, \ldots, a_n$ are the regression coefficients that represent the effects of normalized features, $x_1, x_2, \ldots, x_n$ are normalized explanatory features, $y$ is the DBH of a tree, and $\varepsilon$ is the error term. In the linear model, the target value is expected to be a linear combination of the features. When the number of predictors increases, the estimates of the LM regression coefficients may be unstable and have high standard errors, especially in small datasets.

The LMR method addresses the harmful influence of high data dimensionality and multicollinearity on the robustness of the linear model by imposing a penalty on large absolute values of the coefficients. The regularization term of LMR is the L2-norm of the coefficient vector, which reduces the impact of redundant variables by shrinking their coefficients while increasing the generalization ability of the linear model [15].

The loss function of LMR can be expressed as

$$\min \frac{1}{2n} \sum_{i=1}^{n} (y_i - \hat{y}_i)^2 + \lambda \sum_{i=1}^{m} a_j^2 \tag{2}$$

where $y_i$ is the observed value and $\hat{y}_i$ is the predicted value for tree $i$, $a_j$ is the regression coefficient, $\lambda$ is the regularization coefficient, $n$ is the number of observations, and $m$ is the number of regression coefficients.

The minimized loss function is the mean of squared errors plus the sum of squared regression coefficients multiplied by a hyper-parameter ($\lambda$). Regularization restricts model complexity and decreases the likelihood of overfitting. The hyper-parameter defines the importance of aiming at small absolute parameter values. The hyper-parameter was "tuned", which means that a value resulting in the best fitting statistics in cross-validation was searched.

SVR is a nonparametric machine learning regression algorithm, which is an extension of support vector machines [30]. The loss function of SVR can be expressed as

$$\min \frac{1}{2} \sum_{i=1}^{m} a_j^2 + C \sum_{i=1}^{n} (L_i + L_i^*) \tag{3}$$

subject to

$$f(x_i) - y_i \leq \varepsilon + L_i \tag{4}$$

$$y_i - f(x_i) \leq \varepsilon + L_i^* \tag{5}$$

$$L_i \geq 0, \ L_i^* \geq 0 \ i = 1, 2, \ldots, n \tag{6}$$

where $x_i$ is a vector of predictor variables for tree $i$, $a_j$ is regression coefficient, $y_i$ is the observed DBH and $f(x_i)$ is the predicted DBH, $L_i$ and $L_i^*$ are the relaxation variables, $C$ is the regularization coefficient, $\varepsilon$ is the allowed error, $n$ is the number of observations, and $m$ is the number of predictors. SVR allows predictions to deviate from the measured value by a small amount ($\varepsilon$). Deviations larger than $\varepsilon$ are penalized.

SVR transforms nonlinear regression into linear regression by mapping the input data into a high-dimensional feature space through a kernel function [31]. In this study, we used the radial basis function (RBF) as the kernel function of SVR. The radial basis function can be expressed as

$$K(x,z) = \exp^{(-r\|x-z\|^2)} \tag{7}$$

subject to

$$r > 0 \tag{8}$$

where $x$ is the vector of predictor variables, $z$ is a high-dimensional feature vector which can be expressed as a new data distribution by mapping $x$ into a higher dimensional feature space, and $r$ is the parameter of the radial basis function. The larger the parameter $r$ is, the faster the value of the feature decreases.

The advantages of SVR include efficiency in high-dimensional data, robustness, the possibility to control the effect of outliers, and a good generalization ability [32]. The kernel parameter $r$ of the RBF controls the influence of a single observation and establishes a cost function to mitigate the impact of outliers [31]. Two of the hyper-parameters of SVR were tuned in this study: the cost of violating the restrictions ($C$) and the kernel parameter of the radial basis function (r) [33].

RF is a machine learning method that combines the predictions of several classification and regression trees (CART) to improve generalizability and robustness over a single estimator [34]. CART is trained to predict the value of the target variable through simple decision rules inferred from the input data. A simple tree usually consists of a set of constants that are used sequentially to provide the estimate. The prediction of the RF method is the averaged prediction of several CART estimators. Each estimator is generated by taking a bootstrap sample from the training dataset, and randomly selecting a specified number of features as predictive variables [35]. In this study, two hyper-parameters of the RF method were tuned: the number of trees that were used to provide the average estimate (*ntree*) and the number of predictor variables randomly sampled as candidates at each split (*mtry*) [29].

ANN is a supervised learning algorithm based on multi-layer perceptron (MLP) networks. ANN has already been widely used in forestry surveys and parameter estimation [36]. The advantages of ANN include the capability to learn nonlinear models, learn in real-time (on-line learning) and the use several neurons to model a nonlinear relationship in parallel [37].

ANN consists of two phases: the learning phase and the prediction phase. The learning phase finds the rules between the input layer and hidden layers, as well as hidden layers and the output layer, by establishing a weight matrix between the neurons. A neuron calculates an input signal value as a weighted linear combination of the predictors (Figure 5). An activation function converts this value to an output signal ranging from 0 to 1. This is the hidden layer value produced by a neuron. Through iteratively modifying the weight matrix, the ANN finds an estimator that minimizes the loss function.

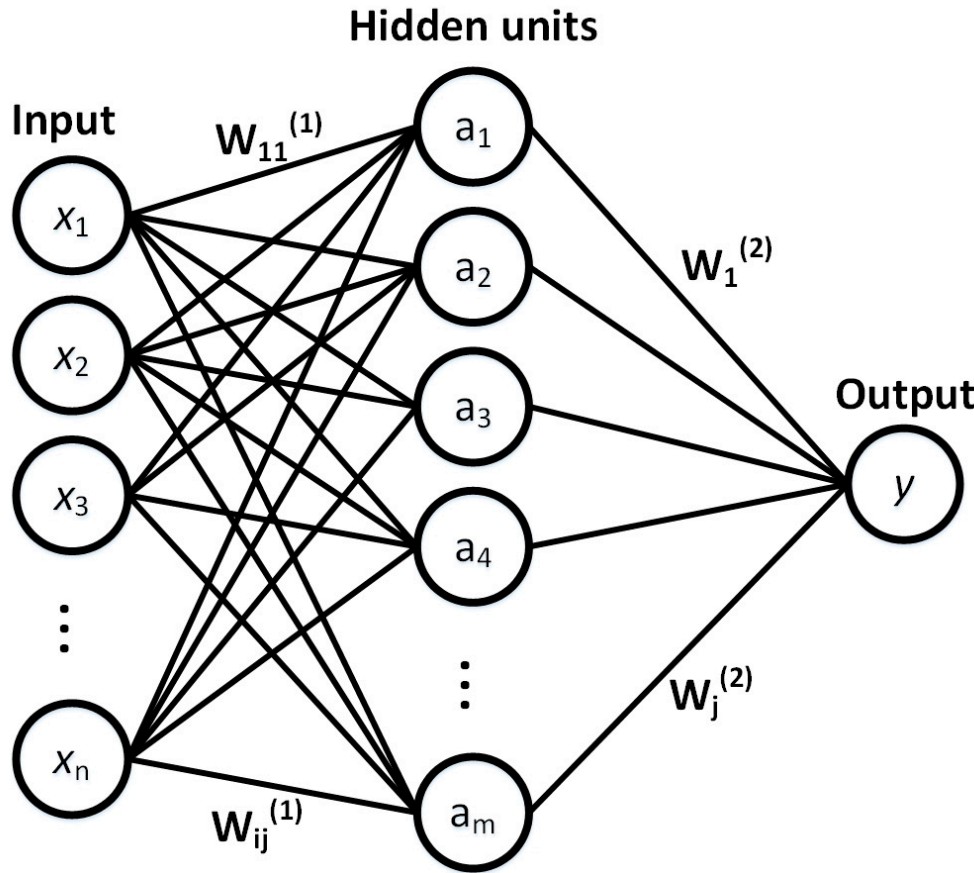

**Figure 5.** An overview of the artificial neural network (ANN) method. Symbol *x* represents an input feature, $W_{ij}^{(1)}$ are the weights of predictors (normalized LiDAR metrics) to calculate intermediate values, $a_j$ is the intermediate value (hidden layer value) of neuron *j*, obtained from the activation function, $W_j^{(2)}$ are the weights of the neurons in the calculation of the prediction, and *y* is the predicted DBH value. In this ANN, there is one hidden layer.

The updated values of the weights are calculated as follows:

$$w' = w - n\frac{\partial LOSS}{\partial w} \tag{9}$$

where $w$ is the weight of a predictor, $w'$ is the updated weight, $\eta$ is the learning rate that controls the step size of the updating process, and *LOSS* is the loss function.

In this study, the loss function minimized the root mean square error. The number of hidden layers was set to two. Three hyper-parameters were tuned: type of the activation function, number of neurons in the hidden layer (*size*), and the weight decay rate ($\eta$). The tested activation functions were logistic function, hyperbolic tangent function, and the rectified linear unit function. The rectified linear function converts negative input values into zero output values and uses a linear relationship between positive input values and the output value.

KNN is a nonparametric imputation method, which is based on the similarity of the LiDAR metrics calculated for a subject tree and field-measured trees. In this study, the Mahalanobis distance was used to calculate the similarity of two trees (a subject tree and its neighbors) [21]. The Mahalanobis distance can be expressed as

$$D = \sqrt{(x - y)^T \sum{}^{-1}(x - y)} \tag{10}$$

where $D$ is the Mahalanobis distance between observed values $x$ and $y$ and $\sum$ is the covariance matrix. The KNN method produces a prediction for the subject tree as a weighted average of the field-measured DBH of several most similar neighbors. The weights of the neighbors were based on the inverse distances of the neighbors from the subject tree. One hyper-parameter of the KNN method was tuned: the number of nearest neighbors that were used to compute the prediction ($k$).

### 2.8. Feature Selection and Hyper-Parameter Tuning

The feature selection and hyper-parameter tuning consisted of the following steps:

1. Select the optimal feature subset using the initial (first repetition) or tuned (later repetitions) hyper-parameters and 10-fold cross-validation.
2. Tune the hyper-parameters by using 10-fold cross-validation and the feature subset found in Step 1.
3. Repeat Steps 1 and 2 using tuned hyper-parameters in Step 1 until the feature subset and the hyper-parameters stabilize.

Before the model training process, all LiDAR metrics were normalized (centered on the mean and scaled by the standard deviation) and the dataset was divided into a training set (75%) and testing set (25%) randomly.

The recursive feature elimination (RFE) algorithm was used for feature selection [31]. The RFE is a greedy method for finding the optimal feature subset, which is specific for each method [38]. The optimal feature subset size was based on the performance of the model, which was evaluated by the root mean square error (RMSE) of a 10-fold cross-validation scheme. The training dataset was divided into 10 subsets (Figure 6). One sub-set was used for testing and nine were used for training. This was repeated 10 times, using every time a different training and test dataset. The results are the averages calculated for the 10 training and test sets.

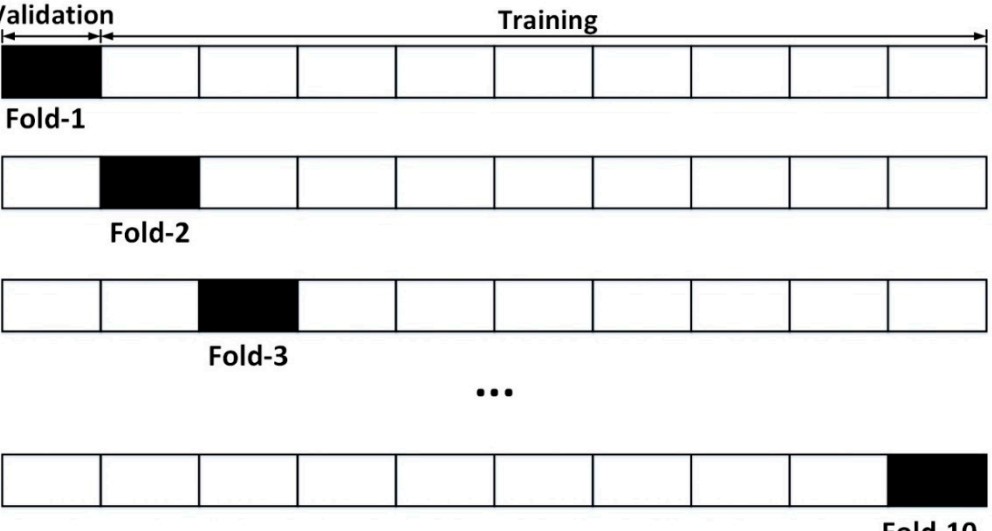

**Figure 6.** The principle of the 10-fold cross-validation. The dataset is divided into 10 parts, of which nine are used for training and one for validation. The subset that is used for validation is different in each of the 10 repeated model fittings and validations.

RFE was used to evaluate the effect of the number of input features on the model performance. The RFE algorithm is based on feature importance, where the relative importance of each metric is obtained by quantifying the increase in the mean square error of the model after the removal of the variable [34]. The optimal feature subset size was the one that minimized the average RMSE of the cross-validation scheme. The values of the hyper-parameters that were used in the feature selection step were taken from previous

studies [8,11,17]: LMR, $\lambda = 0.01$; SVR, $r = 0.0011$, $C = 64$; RF, *ntree* = 400, *mtry* = $m/3$ (*m* is the number of LiDAR-derived metrics used in RF); KNN, $k = 5$; ANN, *size* = 1; $\eta = 0.6$.

After selecting the set of features, a grid-search was used to determine the best values for the hyper-parameters. This was also done by using 10-fold cross-validation; the optimal parameters were those that resulted in the lowest mean RMSE. The hyper-parameter values tested for different algorithms are listed in Table 4.

**Table 4.** Tested hyper-parameters for each machine learning algorithm.

| Model | Hyper-Parameter Values |
|---|---|
| Linear Model (LM) | - |
| Linear Model with Ridge Regularization (LMR) | $\lambda = 0.0001, 0.003, 0.001, 0.003, 0.01, 0.03, 0.1$ |
| Random Forest (RF) | *ntree* = 50, 100, 150, . . . , 300<br>*mtry* = $n \times m$ ($n = 0.1, 0.2, 0.3, \ldots, 0.9$) |
| Support Vector Regression (SVR) | $C = 1, 3, 6, 10, 30, 60, 100$<br>$r = 0.0001, 0.003, 0.001, 0.003, 0.01, 0.03, 0.1$ |
| K-Nearest Neighbors (KNN) | $k = 1, 2, 3, \ldots, 10$ |
| Artificial Neural Networks (ANN) | activation function: {*logistic, tanh, relu*}<br>*size* = 1, 2, 3, . . . , 15<br>$\eta = 0.1, 0.2, 0.3, \ldots, 1$ |

Note: *m* is the number of LiDAR-derived metrics used in RF, *logistic* is the logistic sigmoid function, *tanh* is the hyperbolic tangent function, and *relu* is the rectified linear unit function.

### 2.9. Model Validation

The performance of the DBH prediction methods (LM, LMR, SVR, RF, ANN, and KNN) was assessed in terms of coefficient of determination ($R^2$), square root of mean squared error (RMSE, in cm), relative RMSE (rRMSE, %), and mean error (BIAS, in cm) [39]:

$$R^2 = 1 - \frac{\sum_{i=1}^{n}(y_i - \hat{y}_i)^2}{\sum_{i=1}^{n}\left(y_i - \bar{y}\right)^2} \tag{11}$$

$$RMSE = \sqrt{\frac{\sum_{i=1}^{n}(y_i - \hat{y}_i)^2}{n}} \tag{12}$$

$$rRMSE = \frac{RMSE}{\bar{y}} \tag{13}$$

$$RIAS = \frac{\sum_{i=1}^{n}(y_i - \hat{y}_i)}{n} \tag{14}$$

where $y_i$ represents the observed DBH value of the tree *i*, $\hat{y}_i$ is predicted value of tree *i*, $\bar{y}$ is the observed mean value, and *n* is the number of observations.

### 3. Results

#### 3.1. The Best LiDAR Metrics for Predicting DBH

The RMSEs of most of the DBH prediction methods were not much affected by the number of input features after about 8 variables (Figure 7). The average RMSE stabilized after reaching the best accuracy, except LM where a high number of predictors most likely resulted in overfitting, decreasing model performance in independent test data. The LMR method mitigated the overfitting problem of LM by using regularization. The ANN method was also negatively affected by a high number of input variables, becoming unstable beyond the best number of variables. Since ANN was sensitive to hyper-parameters, the overfitting pattern shown in Figure 7 may be explained by the fact that the hyper-parameters were not

tuned simultaneously with feature selection. Figure 7 indicates that four to ten predictor variables are needed to predict DBH.

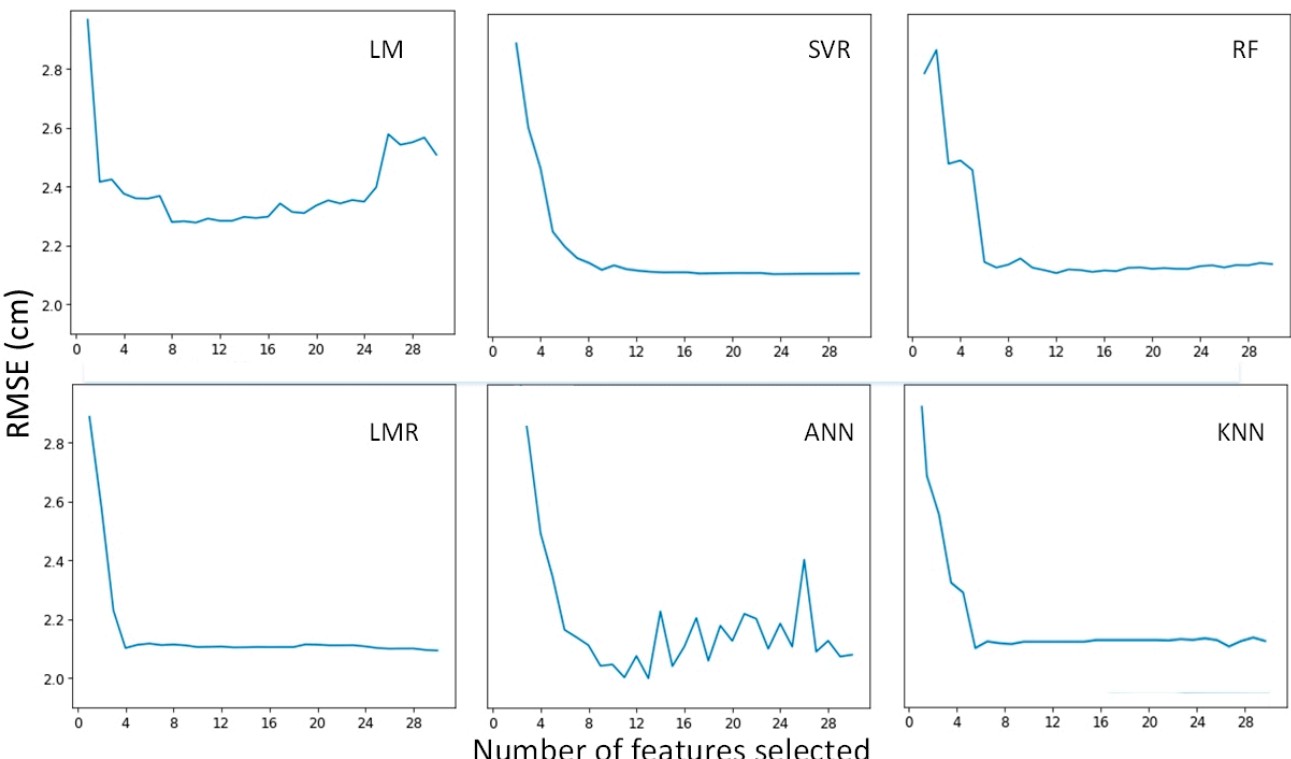

**Figure 7.** Impact of the number of input features on the average of cross-validated RMSE for different prediction algorithms.

The most important LiDAR-derived metrics for estimating DBH were partly the same for the different methods (Figure 8). $H_{max}^{T}$ (maximum point cloud elevation within the crown area of an individual tree) or its transformation $(H_{max}^{T})^2$ usually had the greatest relative importance.

The LiDAR metrics used in the LM regression methods differed most from the metrics used in the other methods. The Pearson's correlation coefficients between the predictor variables that were used in LM were greater than 0.75. The variance inflation factor of LM was greater than 10, which indicates high multicollinearity among the independent variables selected by RFE for the LM.

### 3.2. Performance of the Prediction Methods

The DBH was generally predicted well in the 10-fold cross-validation (Table 5). The prediction accuracy of LMR was close to the nonparametric machine learning techniques suggesting that the regularization of LMR improved the robustness of the model.

The average rRMSE of the nonparametric machine learning methods, SVR, RF, KNN, and ANN was 14.11%, 14.43%, 12.75%, and 11.75%, respectively, which are smaller than obtained for the LM method. The ANN method had the lowest average of RMSE calculated by cross-validation.

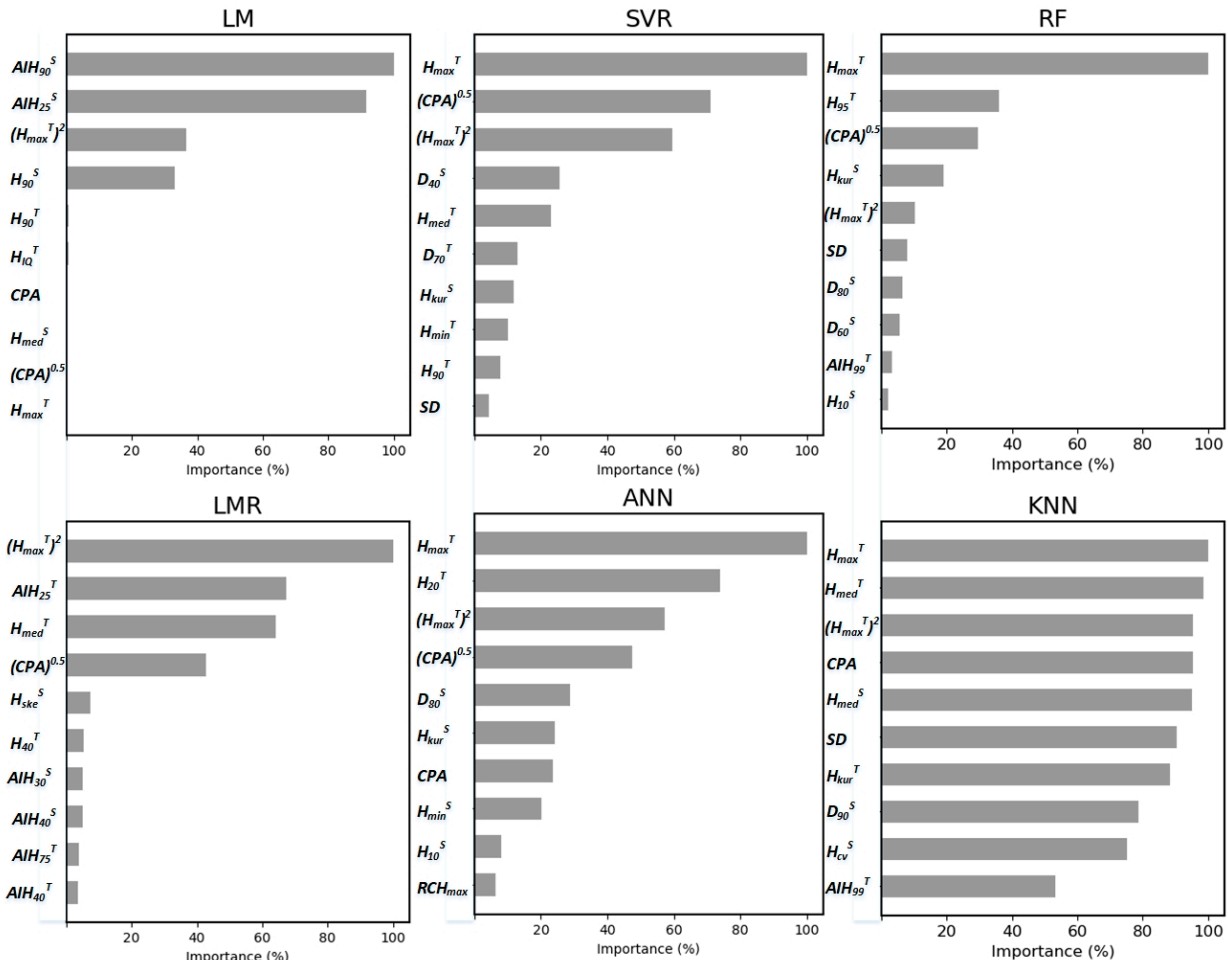

**Figure 8.** The relative importance of the 10 most important metrics for each DBH prediction method. LM used only used seven metrics to predict the DBH of individual trees. A description of the LiDAR-derived metrics is given in Table 3.

**Table 5.** Summary of hyper-parameter tuning and accuracy assessment results in cross-validation for different regression algorithms.

| Model | Hyper-Parameter Values | Statistics | 10-Fold Cross-Validation | | |
|---|---|---|---|---|---|
| | | | $R^2$ | RMSE (cm) | rRMSE (%) |
| LM | - | Mean | 0.82 | 2.56 | 15.59 |
| | | Std Dev | 0.01 | 0.15 | 0.93 |
| LMR | $\lambda = 0.003$ | Mean | 0.83 | 2.09 | 14.15 |
| | | Std Dev | 0.01 | 0.16 | 1.00 |
| SVR | $C = 30$ $r = 0.001$ | Mean | 0.84 | 1.98 | 14.11 |
| | | Std Dev | 0.01 | 0.17 | 1.02 |
| RF | $ntree = 100$ $mtry = 6$ | Mean | 0.81 | 2.36 | 14.43 |
| | | Std Dev | 0.01 | 0.12 | 0.09 |
| KNN | $k = 9$ | Mean | 0.83 | 2.09 | 12.75 |
| | | Std Dev | 0.02 | 0.16 | 0.99 |
| ANN | $size = 10$ $\eta = 0.8$ activation function = *logistic* | Mean | 0.84 | 1.92 | 11.75 |
| | | Std Dev | 0.02 | 0.16 | 1.00 |

Note: Mean is the average value in the 10-fold cross-validation scheme and Std Dev is the standard deviation of the 10 cross-validation values.

### 3.3. Performance of the Prediction Methods in Independent Validation Data

To compare the prediction and generalization ability, we calculated the predicted DBHs for the independent validation dataset (25% of the data, see Figure 2). This validation dataset was not used for feature selection, hyper-parameter tuning, and model fitting. In LM and LMR, the minimum predictions were negative, i.e., these algorithms were not able to logically predict the diameters of very small trees (Table 6).

**Table 6.** Summary of statistical results of the predicted DBHs for the independent validation dataset calculated using six different methods.

| Models | Std Dev | Minimum | Maximum | Range | RMSE | rRMSE | BIAS |
|--------|---------|---------|---------|-------|------|-------|------|
| LM | 7.66 | −0.84 | 38.91 | 39.76 | 2.68 | 15.76 | −0.01 |
| LMR | 7.70 | −1.38 | 38.90 | 39.26 | 2.25 | 13.23 | −0.04 |
| SVR | 7.67 | 3.82 | 36.37 | 32.54 | 2.22 | 13.06 | 0.02 |
| RF | 7.36 | 6.91 | 28.68 | 21.77 | 2.53 | 14.88 | 0.02 |
| KNN | 7.54 | 5.53 | 34.51 | 28.97 | 2.19 | 12.80 | −0.06 |
| ANN | 7.71 | 5.20 | 32.57 | 27.37 | 2.11 | 12.41 | −0.01 |
| Field-measured DBH | 7.98 | 4.50 | 37.60 | 33.10 | - | - | - |

Note: Std Dev = Standard deviation.

Table 6 indicates that the predicted DBH values of the nonparametric machine learning algorithms were close to the field-measured diameters. However, in RF and ANN, the maximum predicted values were smaller than the largest measured values, which means that the RF and ANN algorithms were not able to properly predict the DBH of the largest trees. When looking at all the statistics in Table 6, it can be concluded that the predictions of SVR best reflect the range and the distribution of the measured breast height diameters.

The LM and RF methods had a wider scatter of the predicted values compared to the other algorithms (Figure 9). In addition, the RF systematically predicted the diameters of the largest trees too small. To some extent, this was also true for the ANN method. Otherwise, the scatterplots of the DBH prediction methods were reasonably close to the 1:1 trend line (red line in Figure 9).

The prediction results of the six different algorithms were assessed for three DBH ranges (Table 7). The rRMSE values were the worst for the DBH range less than 12 cm. Although the ANN provided the best overall estimation results (Table 6), SVR provided the best estimation (smallest RMSE and bias) for the DBH range greater than 30 cm. When looking at all the three diameter ranges, the SVR method had the best prediction accuracy. ANN performed slightly better than LMR and KNN.

**Table 7.** Summary of accuracy assessment results for the validation dataset for different DBH ranges. Positive bias indicates underestimation.

| Model | Statistics | DBH Ranges (cm) | | |
|-------|-----------|------|------|------|
| | | <12 | 12–23 | >23 |
| LM | BIAS | −0.44 | −0.50 | 1.28 |
| | RMSE | 2.28 | 2.28 | 2.75 |
| | rRMSE | 27.42 | 12.62 | 10.22 |
| LMR | BIAS | −0.37 | −0.55 | 1.18 |
| | RMSE | 2.22 | 2.24 | 2.54 |
| | rRMSE | 26.32 | 12.39 | 9.46 |
| SVR | BIAS | −0.37 | −0.53 | 1.05 |
| | RMSE | 2.12 | 2.18 | 2.48 |
| | rRMSE | 25.59 | 12.01 | 9.18 |

**Table 7.** *Cont.*

| Model | Statistics | DBH Ranges (cm) | | |
| --- | --- | --- | --- | --- |
| | | **<12** | **12–23** | **>23** |
| RF | BIAS | −0.63 | −0.48 | 1.59 |
| | RMSE | 2.41 | 2.37 | 2.94 |
| | rRMSE | 28.88 | 13.09 | 10.94 |
| KNN | BIAS | −0.39 | −0.74 | 1.19 |
| | RMSE | 1.88 | 2.28 | 2.62 |
| | rRMSE | 22.59 | 12.57 | 9.74 |
| ANN | BIAS | −0.01 | −0.48 | 1.39 |
| | RMSE | 1.92 | 2.19 | 2.64 |
| | rRMSE | 23.18 | 12.11 | 9.75 |

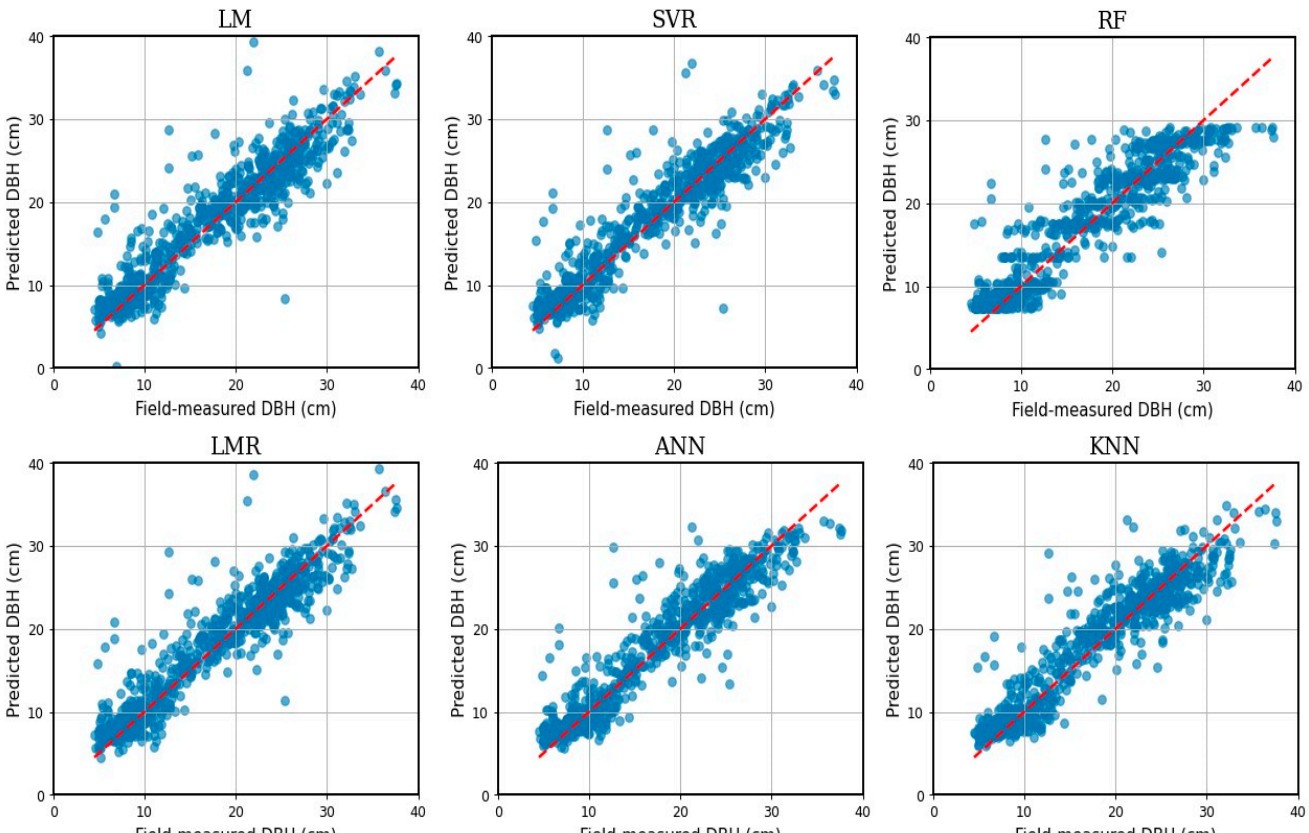

**Figure 9.** The predicted vs. field-measured diameter at breast height (DBH) in the independent validation data for six different prediction algorithms. The red line is the 1:1 trend line.

## 4. Discussion

This study compared the performance of six different methods to predict individual tree diameter from UAVLS data. The average point density of the data was approximately 140 pts·m$^{-2}$, which was sufficient for delineating the crowns of individual trees. Metrics calculated on two scales (tree and stand) were used to predict individual tree diameter.

Previous studies have shown that the use of random stand effects improves the models for predicting tree height from DBH, or DBH from tree height [2,12]. Since the effect of stand variables on the relationship between DBH and tree height is often systematic, it can also be modeled through fixed model parameters. Previous studies show, for example, that the height of a tree with a given DBH increases with increasing stand age or mean height [19]. For example, Hao et al. (2021) used both tree and plot level LiDAR metrics as predictor variables in a nonlinear mixed-effects model, which also included random site

effects [11]. This significantly improved the model compared to the use of fixed tree effects and random site effects. In this study, the set of the potential predictors of individual tree diameter included both tree and stand-level LiDAR metrics, which allowed the relationship between tree height and DBH to change along changing stand conditions.

In the present study, the conventional linear regression model was the poorest of the six tested prediction algorithms. This was partly because the method was unable to address nonlinear relationships, and partly because of the low robustness of the method in the presence of multicollinearity. It might also be that the method used to select the predictors was not ideal for LM because the selected predictors were highly correlated. The lacking ability to model nonlinear relationships was mitigated by using two transformed variables, which linearized the relationship between tree height and DBH, and between crown projection area and DBH.

The transformed variables were selected as predictors also in the other approaches (Figure 8), which suggests that several of the methods analyzed in this study may not be ideal to properly model nonlinear relationships. Common to all methods was that the diameter of medium-sized trees was overestimated and the diameter of the largest trees was underestimated. When the six algorithms analyzed in this study were applied without the transformed variables, the statistics calculated for the independent validation dataset (not shown) were always worse than obtained with transformed variables. The tendency of overestimating the DBH of medium-sized trees and underestimating the DBH of large trees was stronger when transformed variables were not used. On the other hand, the cross-validation results calculated for the training dataset did not always improve when transformed variables were used as additional predictors.

The linear model with regularization (LMR) resulted in a robust model, mitigating the overfitting problem of the LM (Figure 7). The performance statistics calculated from the independent validation dataset were also better for LMR than LM. They were also better than obtained for the random forest method and competitive with the ANN and KNN methods.

The nonparametric approaches do not require a priori specification of the functional form of the relationship between the predictor variables and the response variable [40]. Of the nonparametric machine learning algorithms, ANN, KNN, and SVR had the smallest average RMSE in the 10-fold cross-validation based on the training dataset. However, KNN and ANN underestimated the diameters of the largest trees, which is a serious problem in the calculation of stand volume or biomass, for example. The ANN and SVR showed good performance in the 10-fold cross-validation, which is in line with previous studies [17]. However, ANN had also a serious shortcoming problem as it could be weak in predicting the DBHs of the very largest trees. The time needed to tune the hyper-parameters was much longer for ANN than the other methods.

KNN differs from the other methods in such a way that the predicted diameter is obtained as the mean of the field-measured diameters of a few trees that are similar to it in terms of LiDAR metrics. Therefore, the method needs a database of field-measured diameters when it is used in prediction, which may restrict its usability. In addition, the method cannot predict diameters larger than the largest field-measured diameters, making it difficult to apply the model in a new area, or a few years after field-data collection. The advantage of the method is that several tree-level variables can be imputed to the LiDAR-detected trees without the need to develop separate models for all variables.

The RF algorithm had the poorest performance of the six tested algorithms according to the statistics calculated for the independent validation dataset. A serious shortcoming of the method was its inability to predict small and large tree diameters. The problem may be related to the number of decision nodes used in the decision trees of the RF method. Adding more nodes to classify the large trees in more fine-grained DBH classes would most probably mitigate the problem.

The use of the RFE method for variable selection reduced the number of predictors for reducing the complexity of the model. The RFE method seldom selected intensity-related

LiDAR metrics as predictor variables. Only ANN used a few intensity-related metrics to predict DBH, which however had lower relative importance than the height-related metrics.

All of the nonparametric machine learning methods identified the maximum value of the individual tree point heights ($H_{max}{}^T$) or its transformation as the most important predictor variable for modeling DBH, which is logical as the $H_{max}{}^T$ represents the total height of trees. The result is in line with previous studies [17].

Our study indicated that the relative prediction accuracy was the lowest for small trees. The trees with small DBH values usually represent young stands. Young stands are often dense and the tree crowns form a rather continuous surface, which may cause problems in individual tree detection. The difficulty in predicting the diameters of small trees agrees with the conclusions of previous studies [11].

Failures to detect trees or interpreting tree groups as one tree are most probably common when the forest stand is dense [41]. Previous studies have analyzed the influence of individual tree segmentation methods on the estimation of vegetation characteristics with LiDAR data. It has been found that dense young forests are the most prone to segmentation errors, affecting the overall accuracy of DBH estimation [11,42]. In young stands, there may be many overlapping crowns and the crowns often form a rather uniform canopy surface. This makes it difficult for the watershed segmentation algorithm to detect individual tree crowns. Using a higher value for the crown base height threshold might mitigate the problem to some extent, but it would not eliminate it. Therefore, better individual tree delineation approaches are called for in dense young forests.

The results of the present study were hardly affected by the individual tree segmentation errors because only matched trees were used in the analyses. However, when individual trees detected from LiDAR data are used in forest planning, which seems to be an increasing trend, non-matched trees cannot be omitted. There are already several studies on optimizing forest management at the tree level based on individual tree detection and airborne laser scanning data [15,43,44].

Due to the difficulties to detect all trees of all stands, there is a need to develop new types of methods for tree-level planning, such as the hybrid method discussed in Sun et al. (2022). The data for such methods can be produced by the combined use individual tree delineation (ITD) and area-based approaches (ABA), where the latter produces a complete diameter distribution of trees, of which a part can be detected individually [45]. The idea of the hybrid method for management optimization is to use a different decision rule for those trees that are detected individually and trees that are known only via the diameter distribution.

Using individual trees in forest planning can use also the information on tree locations. Knowing the tree locations would make it possible to use distance-dependent growth models, analyze spatial problems, or optimize the locations of harvest roads, for example.

Although LiDAR has been recognized as a reliable technology to provide detailed data from forests in three dimensions, it has a restricted spectral resolution, generally covering only a single spectral range in the near-infrared region [4]. Therefore, future studies should focus on integrating multi-source remote sensing data for improved prediction of tree diameters and other characteristics of trees and stands. Integrated use of multi-source information can lead to improved prediction accuracy, especially for mixed forests, where the tree species of each detected tree needs to be predicted.

## 5. Conclusions

The study showed that several nonparametric methods can be used to develop models for predicting individual tree diameter from LiDAR data. In addition, regularization methods can be used to increase the robustness of the ordinary linear regression model (LM) and decrease the overfitting problem of this model in the case of a high number of mutually correlated potential predictors. All the methods tested as alternatives for the ordinary linear regression model performed better than the LM. However, the improvements were not drastic, most probably because of the strong correlation between LiDAR-estimated canopy

height and tree diameter. In addition, all the six prediction methods had the drawback that the diameters of the largest trees were underestimated, which might be because of the methods' inability to properly model nonlinear relationships. This shortcoming could be at least partly overcome by using different transformations of the potential predictors. The underestimation of large diameters was the most severe with the random forest method. The support vector regression was evaluated to be the best method for predicting individual tree diameters from LiDAR data. The performances of the linear model with ridge regularization, the k-nearest neighbors imputation, and the artificial neural network method were close to each other. The benefits of the methods analyzed in this study might be larger for predicted variables other than tree diameter, for instance, crown length, crown volume, or tree biomass. The benefits of the methods tested in this study can be expected to be the highest in the case of little prior knowledge on the relationships between the variables, a high number of potential predictors, and strong correlations between the potential predictors.

**Author Contributions:** Conceptualization, T.P.; methodology, T.P. and Y.S.; formal analysis, T.P. and Y.S.; data curation, X.J. and F.L.; writing—original draft preparation, T.P. and Y.S.; writing—review and editing, X.J. and F.L.; supervision, X.J. and F.L.; project administration, X.J.; funding acquisition, F.L. All authors have read and agreed to the published version of the manuscript.

**Funding:** This research was supported by Natural Science Foundation of China (U21A20244) and (32071758) and the Fundamental Research Funds for the Central Universities of China (No. 2572020BA01).

**Institutional Review Board Statement:** Not applicable.

**Informed Consent Statement:** Not applicable.

**Data Availability Statement:** Not applicable.

**Acknowledgments:** The authors would like to thank the faculty and students of the Department of Forest Management, Northeast Forestry University (NEFU), China, who collected and provided the data for this study.

**Conflicts of Interest:** The authors declare no conflict of interest.

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
