# Peer review of "Predicting Individual Tree Diameter of Larch (Larix olgensis) from UAV-LiDAR Data Using Six Different Algorithms"

_remotesensing, doi:10.3390/rs14051125_

Round 1

Reviewer 1 Report

Overall, the article is well written; however, it does not present anything new to the area, it tests DBH estimation methods already known in the literature, but it has the qualities to be accepted in the Remote Sensing journal.
As the main weakness, I point out that the validation of the height of the trees and the stand-by LiDAR was not carried out. As the DBH was estimated from the LiDAR height, the error of this estimate should have been evaluated. So, what was the error in estimating the height of the trees and the stand?

Resume
Line 21 - What metrics? Traditional metrics based on height information.

1.Introduction
What is the paper's contribution? This should be highlighted in the introduction.

2. Materials and Methods
2.1 Study sites and field data
Insert a table with the main dendrometric characteristics (min, max, mean, SD) (DBH and total height ) of the trees estimated in the plots.

Line 117 - Set DBH.
Line 149 - Explain the reason for using Gaussian-smoothing filter. Cite some references that use this filter to remove the noise data points.
Line 150 - What parameters are used in cloth simulation filtering (sloop_smooth , class_threshold cloth_resolution , rigidity , iterations, time_step ) ?
Lines 149 - 152 - What software was used?
Line 158 -Individual tree delineation
Enter the omission and commission errors values.

Line 160 - What parameters are used in the watershed segmentation algorithm?
Line 169 - Insert the number of trees correctly identified.
Figure 4 - Insert legends for plots and Stand delineation.
Line 196-197 - Indicate, according to Table 2, which are the density-related metrics, canopy volume metrics, and topography-related metrics.
Did the trees in the plot that are within a given stand receive the same metric values ​​extracted for the stand? This is not clear in the methodology, explain.
Line 208 - Why didn't you perform a transformation on all metrics? And did you check if there was an improvement in the correlation with DBH? Why didn't you make other changes as well? It makes no sense to leave Hmax and (Hmax)² as predictor variables, and they are highly correlated.
Line 219 - There are several methods/techniques of selection of variables in the adjustment of the linear regression model; explain why they were not used in the study.

4.Discussion
Line 488 - The RF method also had problems estimating the smallest DBH; explain why.
Lines 497-500 - The Hmax values ​​estimated by LiDAR were not validated with the field data; therefore, this conclusion can be reached.
Line 506 - What were omission and commission errors?

Reviewer 2 Report

This is a well-written and well-studied manuscript on a very relevant topic: how does the selection of a machine learning algorithm influence the estimation of conifer DBH. I am happy by the study and the results, apart from one minor remark.
It seems that the authors assume that every polygon corresponds to a single tree. They also only use trees that have been identified in the field to associate with a polygon. But as they remark on lines 513-515, in operational use it is necessary to account for polygons that correspond to more than one tree - or to only part of a tree! So I hope they would add a few sentences in the discussion addressing the possible importance and impact this challenge would have on the accuracy of UAV-LiDAR for forest inventory in operational context, without pre-screening polygons.
Besides, Table 5 is awkwardly split on tow pages. But that is all I have to comment. 

Reviewer 3 Report

The article entitled " Predicting individual tree diameter of larch (Larix olgensis)  from UAV-LiDAR data using six different algorithms”, tested methods such as the linear regression model (LM), the model with ridge regularization (LMR), the support vector regression (SVR), the random forest (RF), the artificial neural network (ANN), and the k-nearest neighbors (KNN) to predict the DBH (diameter of the tree stem at the breast height) for larch (Larix olgensis) plantations from Northeast China. The tree-level and stand-level LiDAR metrics were used as potential predictors of DBH.  The research identified SVR as the best method for predicting individual-tree diameters from LiDAR data. The authors observed that the diameters of the largest trees were underestimated especially by the random forest (RF) methodology. The paper is well structured and provides useful information for the readers of the journal. The paper can be published with minor revisions.

  1. Abstract is written well.
  2. The methodology section clearly presents the study sites, the UAV-LS data collection and processing, the tree-level LiDAR metrics extraction with stands and individual trees delineation, prediction methods and data validation.
  3. The results section, concerning the DBH prediction using six different algorithms from UAV- LiDAR data, has been written in a consistent manner and it looks convincing.
  4. The references section cites the latest papers in every context to show the present time relevance of the individual tree diameter prediction. 

Author Response

Several minor improvements were made to the revised version of the  manuscript.

Round 2

Reviewer 1 Report

Congratulations to the authors. The article can be accepted in the manuscript.